
# SORDOR Pulses: Expansion of the Böhlen-Bodenhausen Scheme for Low-Power Broadband Magnetic Resonance

Jens D. Haller[1], David L. Goodwin[1], and Burkhard Luy[1,2]

[1]Institute for Biological Interfaces 4 – Magnetic Resonance, Karlsruhe Institute of Technology (KIT), Karlsruhe, Germany
[2]Institute of Organic Chemistry, Karlsruhe Institute of Technology (KIT), Karlsruhe, Germany

**Correspondence:** David L. Goodwin (david.goodwin@partner.kit.edu) and Burkhard Luy (burkhard.luy@kit.edu)

**Abstract.** A novel type of efficient broadband pulses, called SORDOR, has recently been introduced. In contrast to adiabatic excitation, SORDOR-90 pulses provide effective transverse $90°$ rotations throughout their bandwidth, with a quadratic offset dependence of the phase in the $x, y$-plane. Together with phase-matched SORDOR-180 pulses, this enables the Böhlen-Bodenhausen broadband spectroscopy approach for frequency-swept pulses to be extended to any type of $90°/180°$ pulse-delay
sequence. Example pulse shapes are characterised in theory and experiment and an example application is given with a $^{19}$F-PROJECT experiment for measuring $T_2$-relaxation times without distortions due to J-coupling evolution.

## 1   Introduction

Many magnetic resonance applications require the manipulation of spins over a large bandwidth under severe restrictions concerning available rf-amplitude and/or tolerable rf-energy. With the advent of $1.2$ GHz NMR spectrometers practically all
common heteronuclear experiments require the use of amplitude- and phase-modulated pulses due to the enlarged bandwidths that need to be covered and equally modern pulsed EPR spectroscopy at any field strength benefits substantially by the use of shaped pulses (Doll and Jeschke, 2014; Spindler et al., 2015, 2017; Jeschke, 2019). But even conventional NMR spectroscopy at intermediate field strengths at e.g. 600 MHz gain enormously from shaped pulses in conventional carbon-correlated experiments (Ogura et al., 1996; Hwang et al., 1997; Kupče and Freeman, 1997; Cano et al., 2002; Armstrong et al., 2004; Skinner
et al., 2006; Tzvetkova et al., 2007; Enthart et al., 2008) and even more so with particular nuclei like $^{19}$F, $^{31}$P, $^{15}$N, $^{119}$Sn, $^{195}$Pt (Enders et al., 2014; Power et al., 2016; Lingel et al., 2020).

A multitude of composite and shaped pulses has been designed to cope with the bandwidth problem (Levitt, 1982; Shaka and Freeman, 1983; Warren, 1984; Tycko et al., 1985a; Levitt, 1986; Lurie, 1986; Shaka and Pines, 1987; Freeman et al., 1988; Zax et al., 1988; Ewing et al., 1990; Keniry and Sanctuary, 1992; Hull, 1994; Kobzar et al., 2004, 2008; Ehni and Luy, 2012) with
recent pulse shapes optimised using algorithms derived from optimal control theory (Conolly et al., 1986; Mao et al., 1986; Rosenfeld and Zur, 1996; Skinner et al., 2003; Kobzar et al., 2005; Spindler et al., 2012; Ehni and Luy, 2012, 2013, 2014; Koos et al., 2015, 2017) being close to physical limits (Kobzar et al., 2004, 2008, 2012). Whenever only a single component of magnetisation vectors needs to be transferred, excitation and inversion pulses, as members of *point-to-point* (PP) pulses, provide very efficient solutions (Shaka, 1985; Emsley and Bodenhausen, 1990; Garwood and Ke, 1991; Abramovich and Vega, 1993;





Kupče and Freeman, 1994; Barker et al., 2001; Smith et al., 2001; Cano et al., 2002; Skinner et al., 2003, 2004, 2005, 2006;
Gershenzon et al., 2007; Coote et al., 2021), regardless, only *universal rotation* (UR) pulses can be used as full replacements
of conventional, bandwidth-limited, hard pulses. UR pulses, however, are especially demanding regarding both rf-amplitude
and rf-energy (Tycko et al., 1985a, b; Cho and Pines, 1987; Shaka and Pines, 1987; Levitt, 1988; Garwood and Ke, 1991;
Wimperis, 1991, 1994; Emsley and Bodenhausen, 1992; Poon and Henkelman, 1992, 1995; Abramovich and Vega, 1993; Luy
et al., 2005; Brown, 2011; Anand et al., 2012; Kobzar et al., 2012; Moore et al., 2012; Odedra and Wimperis, 2012; Skinner
et al., 2012; Altenhof et al., 2019).

To reduce such demands, several concepts involving matching pulse shapes have been developed. Possibly the first such
concept based on adiabatic pulses has been reported by Böhlen and Bodenhausen, requiring matched, linear frequency-swept
excitation and inversion pulses (Böhlen et al., 1989, 1990; Burghardt et al., 1990). As the effective pulse phases are matched in
this concept, the inversion pulses act as refocusing (UR-180) pulses with a quadratic offset dependence of the corresponding
rotation axes. Matching UR-90 pulses, however, are not available, rendering the applicability of the concept to special cases. A
more general concept, with the least amount of pulse restrictions, is the COOP concept introduced by Braun and Glaser (Braun
and Glaser, 2010, 2014; Asami et al., 2018). However, pulse shapes introduced so far result in similar restrictions.

Here we propose the recently introduced *second order phase dispersion by optimised rotation* (SORDOR) pulses of Goodwin
et al. (2020) for an extension of the Böhlen-Bodenhausen concept for a widely applicable cooperative broadband shaped pulse
scenario, in which $90°$ universal rotations can also be realised. SORDOR pulses define a new class of low-energy pulse shapes
that cause a defined rotation with constant rotation angle over a specified bandwidth like common UR pulses. Instead of
the constant rotation axes of conventional UR pulses, SORDOR pulses show a quadratic offset-dependent phase-change of
the effective rotation axes in the $x, y$-plane, similar to the adiabatic sweeps in the Böhlen-Bodenhausen concept. If a simple
quadratic phase correction of spectra can be tolerated, SORDOR pulses can be applied as a direct replacement of $90°$ and $180°$
hard pulses. They therefore represent a direct extension of the Böhlen-Bodenhausen concept to the requirements of $90°$-based
mixing. After an introduction of pulse properties, their application in NMR spectroscopy is demonstrated experimentally.

## 2 Theory

SORDOR pulses used here have been optimised as described in reference Goodwin et al. (2020): starting from the identity
matrix, effective propagation for a single spin $1/2$ is calculated and the effective propagator $\mathbf{U}_{\text{eff}}$ of a given pulse is used to
calculate a quality factor $\Phi = \text{Re}\langle \mathbf{U}_{\text{eff}} | \mathbf{U}_T \rangle$ where the target propagator is defined by

$$\mathbf{U}_T = \exp\left(-i\beta[\cos\left(\alpha(\omega_z)\right)\mathbf{I}_x + \sin\left(\alpha(\omega_z)\right)\mathbf{I}_y]\right) \tag{1}$$

with

$$\alpha(\omega_z) = Q\frac{t_p}{2\Omega}\omega_z{}^2 + \alpha_0, \tag{2}$$





where $\beta$ defines the desired effective flip angle, $\alpha$ a phase in the $x,y$-plane, $\omega_z$ the offset, $\mathbf{I}_x$ and $\mathbf{I}_y$ the usual spin operators represented by corresponding Pauli matrices, the desired offset range $\Omega$, the pulse length $t_p$, an arbitrary scaling factor $|Q| < 1$, and an arbitrary constant phase $\alpha_0$.

    It is evident from the target propagator that a uniform rotation angle $\beta$ is targeted, resulting in a pulse class termed *B1 pulses* (not to be mistaken with $B_1$ fields), as defined originally by Levitt (1986). However, in addition to the minimum requirement

of class B1 pulses, an offset-dependent target phase for the effective rotation axes in the $x,y$-plane is applied. Such a phase originally has been introduced with *ICEBERG pulses* of Gershenzon et al. (2008) but, in contrast to the linear ICEBERG phase, the SORDOR target phase varies with a quadratic function of the offset $\omega_z$. Other phase dependent control targets have been investigated more recently by Altenhof et al. (2019) and Coote et al. (2021). The quadratic SORDOR phase resembles the quadratic phase being acquired in linear frequency-swept adiabatic inversion pulses (Baum et al., 1985; Kupče and Freeman,

1995; Garwood and DelaBarre, 2001; Jeschke et al., 2015), which can be used advantageously as will be shown below.

    Using the GRAPE algorithm (Khaneja et al., 2005; de Fouquieres et al., 2011; de Fouquieres, 2012; Goodwin and Kuprov, 2016) with exact gradients (Van Loan, 1978; Goodwin and Kuprov, 2015) as discussed in detail in Goodwin et al. (2020), pulses for $\beta = 90°$ and $\beta = 180°$ were obtained, named SORDOR-90 and SORDOR-180, respectively. While the optimisation of SORDOR-180 pulses turned out to be straightforward, good SORDOR-90 pulses were only obtained after introducing a

morphing procedure in the optimisation algorithm (morphic-GRAPE), in which the scaling factor $Q$ and the pulse length $t_p$ are slowly adjusted from extrema.

    The morphic-GRAPE procedure to produce the SORDOR pulses of Goodwin et al. (2020) used four *directional morphs* to produce high fidelity 90° and 180° pulses. The highest fidelity pulses for a bandwidth of 40 kHz, unsurprisingly, were at the longest durations, $t_p = 450$ μs with rf-amplitudes of 10 kHz. Following on from those optimisations, the SORDOR pulses used

in this work introduce an additional directional morph stage: a *ramping* stage to increase the desired bandwidth to $\Omega = 50$ kHz. To this end, the SORDOR pulses ramp 450 μs pulses from 40 kHz to 50 kHz, over $Q = [0.70, 0.71, ..., 0.85]$ (these $Q$ values are used as the high performance SORDOR pulses lay in this range), in increments of 1 kHz.

    Increasing the desired bandwidth lowers the pulse performance. To gain higher performance, the SORDOR pulses ramped to $\Omega = 50$ kHz are further expanded to longer pulse durations from 450 μs to 750 μs, over the best performing $Q$ from the ramped

stage: $Q = [0.78, 0.79, 0.80, 0.81, 0.82]$. The best performing SORDOR pair, at a nominal pulse duration $t_p = 720$ μs, occured at $Q = 0.80$ and this pulse pair was further optimised for $\pm 5\%$ $B_1$-inhomogeneity over a Guassian distribution of rf-amplitude multipliers.

    The resulting SORDOR-90 and SORDOR-180 pulse pair with identical phase dependence $\alpha(\omega_z)$ is shown in Fig. 1 together with a constant-adiabaticity chirped excitation (Khaneja, 2017; Foroozandeh, 2020) pulse (CA-Chirp-exc) for comparison. The

SORDOR pulse shapes have a constant rf-amplitude of 10 kHz and in both cases a smooth rf-phase, even if the course of the rf-phase looks somewhat erratic. The CA-Chirp-exc pulse, instead, has a lower rf-amplitude of only 2850 Hz and the well-known smooth rf-phase behaviour. The reason for the difference in rf-energy consumption of the SORDOR-90 and CA-Chirp-exc pulse can be seen in the analysis of resulting effective rotations (Fig. 1): while the SORDOR-90 at all offsets essentially resembles pure 90° rotations with rotational axes in the $x,y$-plane, the CA-Chirp-exc pulse resembles a classical PP excitation

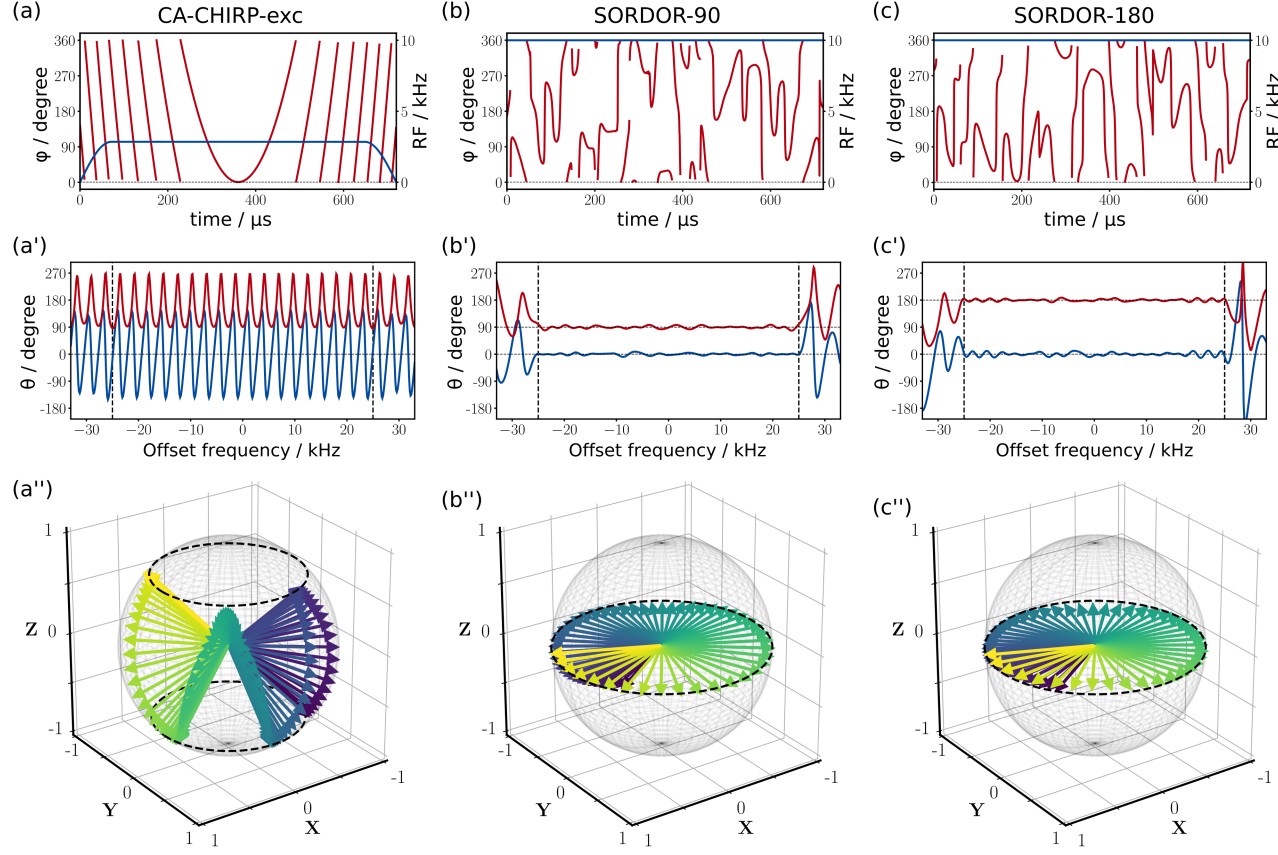

**Figure 1.** Comparison of a constant adiabaticity chirped excitation pulse (CA-Chirp-exc), a SORDOR-90 and the matched SORDOR-180 pulse. (a-c) Pulse rf-amplitudes and phases. (a′-c′) Offset dependencies of the $\sqrt{x^2+y^2}$ (red) and $z$-components (blue) of the effective rotations for the specific pulses. (a″-c″) Visualisations of the normalised rotational axes of the different pulses for a reduced offset region. Note that the Chirp excitation pulse does not perform universal $90°$ rotation while SORDOR-90 does. Please see the SI for an enlarged figure including a Chirp inversion pulse.





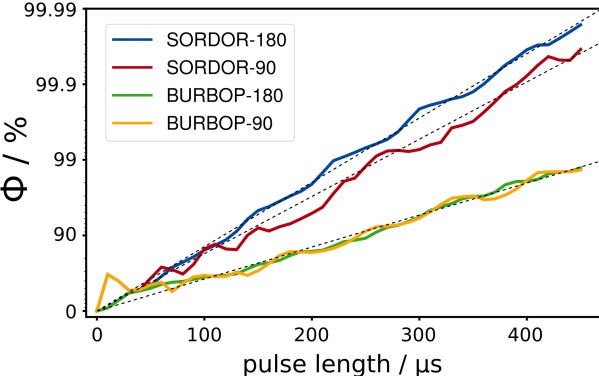

**Figure 2.** Quality factors $\Phi$ for different optimised SORDOR and BURBOP pulse shapes with rf-amplitudes of 10 kHz and optimised bandwidths of 40 kHz. SORDOR-180 pulses of same quality consume approximately half the rf-energy as corresponding BURBOP-180 UR pulses. SORDOR-90 compared to time-optimal BURBOP-90 pulses lead to a reduction in pulse length and rf-energy of approximately a factor 1.8.

pulse with effective rotations in the range of 90°-270° and a periodic elevation of the rotational axes along $z$ with the offset. It is obvious that the adiabatic pulse can only be used to excite a single component, but not as a pulse that produces a defined rotation as required in sequences with coherence transfer involving several Cartesian components. The SORDOR-180 pulse shows the very same rotation behaviour as the SORDOR-90 pulse, just with an effective 180° rotation angle in the $x, y$-plane throughout the optimised offset range. As such it resembles very much the behaviour of an adiabatic inversion pulse of same

duration and rf-amplitude (data shown in SI). However, the SORDOR-180 pulse is directly matched in its phase behaviour to the SORDOR-90 pulse, which for an adiabatic inversion pulse would only accidentally be the case.

A systematic study of SORDOR-90 and SORDOR-180 pulse performances for optimal $Q$-values around 0.75 and a bandwidth of 40 kHz as reported in reference Goodwin et al. (2020) is shown in Fig. 2: with increasing pulse lengths the logarithmic quality factor for pulse performance steadily increases in both cases. As in previous studies exploring the physical

limits (Kobzar et al., 2004, 2008, 2012), the increase does not describe a smooth function, but as a guide to the eye the pulse performance can be roughly described by a linear function (see Fig. 2). Also BURBOP-90 and BURBOP-180 pulses, which can be considered special SORDOR pulses with $Q = 0$, follow a similar curve (Kobzar et al., 2012), but for the same performance approximately twice the amount of rf-energy is needed for the 180° case and about a factor 1.8 for the 90° case. The energy consumption of SORDOR-180 pulses therefore resembles that of time-optimal BIP and BIBOP inversion pulses (Smith

et al., 2001; Kobzar et al., 2004), while the energy consumption of SORDOR-90 pulses lies in between time-optimal BEBOP excitation and BURBOP universal rotation 90° pulses.

The intriguing consequences of the SORDOR-90 and 180 pulse pairs become evident in the frequency sweep pulse concept of Böhlen and Bodenhausen (Böhlen et al., 1989, 1990; Burghardt et al., 1990), which lately has found a revival in EPR spectroscopy (Doll and Jeschke, 2016, 2017a, b). The concept is schematically shown for the perfect echo sequence (Takegoshi





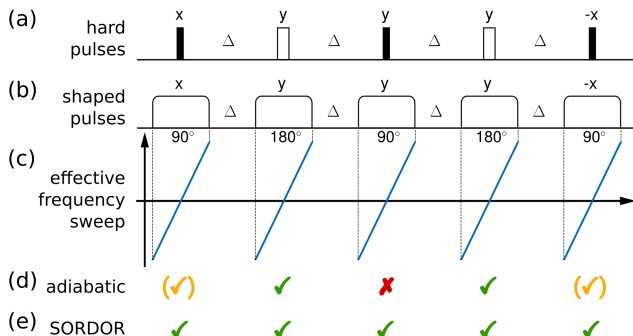

**Figure 3.** Schematic on the applicability of the Böhlen and Bodenhausen concept. (a) hard pulse perfect echo sequence; (b) corresponding shaped pulse scheme; (c) effective frequency sweep of adiabatic and SORDOR pulses with quadratic phase dependence; (d,e) applicability of corresponding adiabatic and SORDOR pulses. Green check-mark: applicable without restriction within the sequence considering that potentially existing coherences have been excited with equally frequency-swept pulses; yellow check-mark: applicable as excitation pulses in this sequence when starting from $\mathbf{I}_z$ polarisation, but not if the full perfect echo sequence is used for coherence transfer; red cross: not at all applicable as a mixing pulse as needed at this position.

et al., 1989) in Fig. 3: A train of $90°$ and $180°$ pulses with delays ensures planar mixing conditions and therefore spin state conservation and inphase to inphase transfer. The sequence works well with hard pulses, but offset limitations sometimes require the application of broadband pulses. With frequency-swept adiabatic pulses, as shown schematically in Fig. 3, in principle any bandwidth can be covered using the fact that even during adiabatic pulses couplings and chemical shifts are evolving effectively with a defined delay, just that the position of the delays is offset-dependent. While this concept using

adiabatic pulses works very well e.g. for a simple excitation and spin echo, the perfect echo requires a defined mixing pulse in the centre with effective $90°$ rotations in the $x,y$-plane. Adiabatic excitation pulses, as shown for CA-Chirp-exc in Fig. 1, cannot fulfil this requirement and the approach is bound to fail. Using the SORDOR-90 and SORDOR-180 pulse pair, instead, retains the Böhlen-Bodenhausen frequency sweep approach fully functional also for the perfect echo. As the SORDOR pulses have an effective quadratic phase-behaviour, they directly behave like linearly swept pulses with well-defined phases, which

either imply the use of a quadratic phase correction (see below) or the application of a SORDOR-180 pulse with twice the rf-amplitude and half the pulse length for phase refocusing (Böhlen et al., 1989, 1990; Burghardt et al., 1990; Doll and Jeschke, 2016). Using the SORDOR pulse pair, the Böhlen-Bodenhausen approach will be applicable in any type of homonuclear $90°/180°$ pulse-delay sequences.

## 3    Experimental

The offset-dependent performance of a single SORDOR-90 pulse and a combination SORDOR-90–$\Delta$–SORDOR-180–$\Delta$ acquisition is shown in Fig. 4. in comparison with spectra acquired using corresponding hard pulses. While a single rectangular





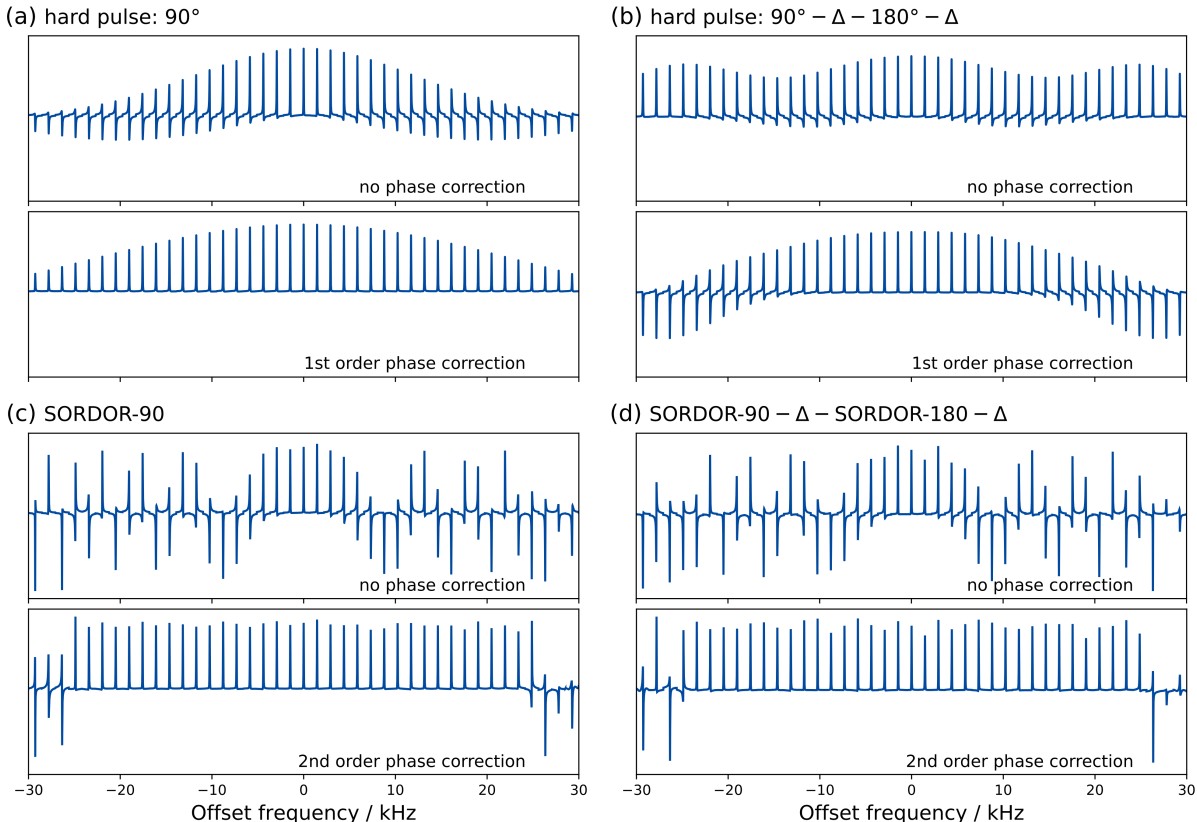

**Figure 4.** Experimental excitation profiles of a hard $90°$ (a) and the SORDOR-90 (c) pulse, as well as a $90°–\Delta–180°–\Delta$ pulse sequence element for hard and SORDOR pulses (b,d). Offset profiles of pulses are shown without phase correction and with linear (first order, (a) and (b)) and quadratic (second order, (c) and (d)) phase correction. The Bruker AU program used for second order phase correction is given in the supporting information.

$90°$ pulse results in a perfectly phaseable spectrum over large bandwidths, the $90°–\Delta–180°–\Delta$ sequence has a clearly reduced bandwidth. Also the signal intensities are rapidly reduced with the offset for the two hard pulse scenarios. The SORDOR-90 pulse, instead, produces a horribly looking offset dependence, which, however, results in nearly constant amplitude signals over the optimised bandwidth if a quadratic phase correction is applied. Equally, the SORDOR pulse combination results in nicely refocused spectra after the second order phase correction.

The applicability of the SORDOR pulse pair is demonstrated in a second example in a $^{19}$F-PROJECT experiment for the measurement of $T_2$-relaxation times without distortions due to coupling evolution (Takegoshi et al., 1989; Aguilar et al., 2012). Corresponding spectra are shown in Fig. 5 for a mixture of $1,1,1,2,3,3-$hexafluoro$-4-$butanol, $1,2-$dichloro$-4-$iodo$-$heptafluoro$-$n and $1,2,3,4-$tetrafluoro$-$salicylic$-$acid in DMSO with $^{19}$F chemical shifts ranging from $-63$ to $-216$ ppm. On the Avance III HD 400 MHz spectrometer used, this corresponds to a bandwidth of 58 kHz, which can be easily covered by the SORDOR

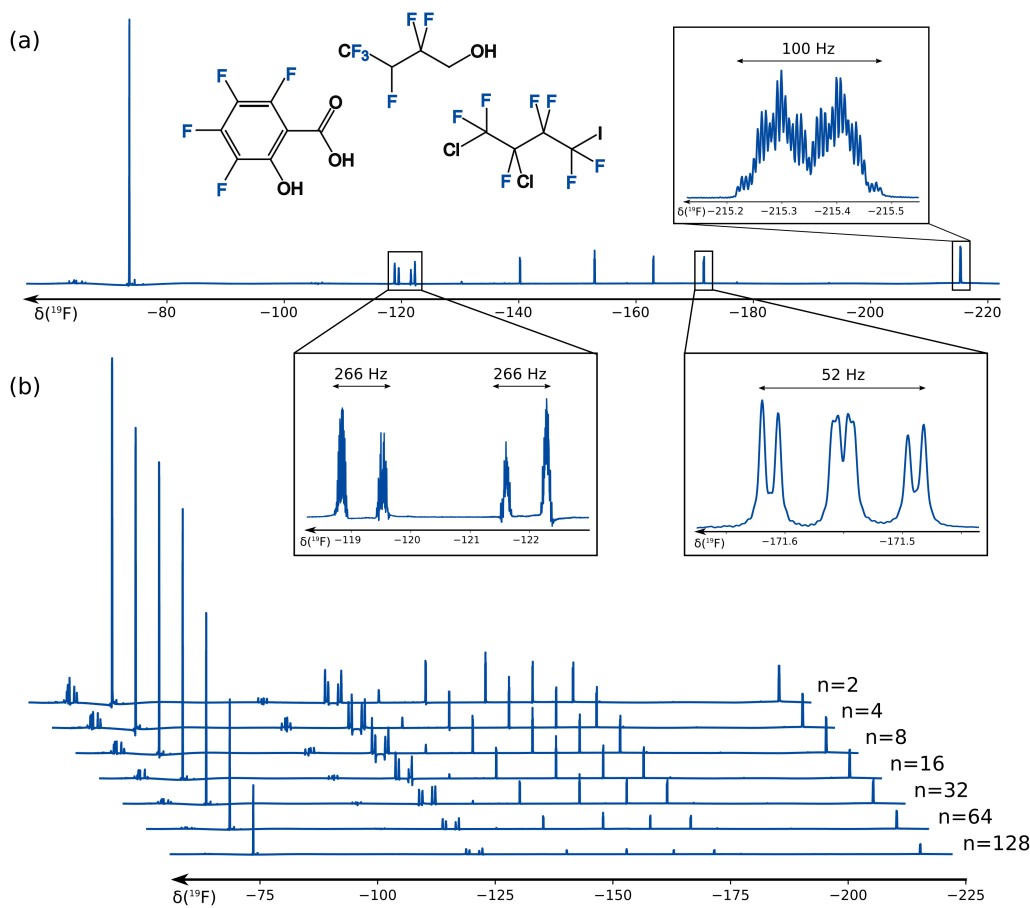

**Figure 5.** $^{19}$F-PROJECT experiment (Aguilar et al., 2012) for $T_2$-relaxation measurement based on multiple SORDOR perfect echoes ($n = 2 - 128$) with $\Delta = 3$ ms and SORDOR pulse lengths $t_{\mathrm{p}}$ of 576 µs each (b). In addition, zooms for several large multiplets for the PROJECT spectrum with $n = 64$ are given (a). Multiplet widths or $^2 J_{\mathrm{FF}}$ coupling constants, respectively, are annotated.



pulse pair with rf-amplitudes of 12.5 kHz. Clearly the relaxation of all signals can be easily followed using up to 128 perfect echoes with delays of $\Delta = 3$ ms and corresponding 576 μs SORDOR-90 and SORDOR-180 pulses, which add with 432 μs (75%) per pulse to the effective delay. Thereby it is important to note that phase distortions due to coupling evolution are ex-

pected for a sum of couplings significantly larger than $1/\Delta$. We therefore had a closer look to multiplets with large couplings that frequently appear in fluorinated compounds. Remarkably, multiplet widths up to 100 Hz still result in pure absorptive PROJECT spectra, demonstrating the robustness of the approach. Only the signals from a $CF_2$ group at 118-123 ppm with a $^2J_{FF}$ coupling of 266 Hz show small distortions, which seem to originate from residual coupling evolution with second order artefacts of the closely resonating signals.

## 145    4    Discussion

Matched SORDOR-90 and SORDOR-180 pulses form a novel class of pulses with uniform rotation angles and a specific quadratic offset dependence of corresponding rotation axes. With such pulses the concept of Böhlen and Bodenhausen can generally be implemented with low rf-energy deposition. In contrast to the original implementation of the concept with adiabatic excitation and inversion pulses, the SORDOR implementation allows the full replacement of 90° and 180° hard pulses in

pulse-delay sequences, including 90° mixing pulses.

As the most simple application, the concept allows the acquisition of 1D-type spectra with only an additional quadratic phase correction. The AU program for Bruker spectrometers used for Fig. 4 to perform a quadratic phase correction is given in the supporting information. This quadratic phase correction is always the same for a given pulse and can be determined once and then applied in all types of experiment with excitation using the same SORDOR-90 pulse.

In more complex pulse-delay experiments, like the PROJECT sequence shown in Fig. 5 for a $^{19}F$ application, hard 90° and 180° pulses can directly be replaced by SORDOR-90 and SORDOR-180 pulses. If the last part of the sequence before acquisition consists of a $\Delta$-180°-$\Delta$ element, the quadratic phase correction of the acquired FID may be compensated by using a SORDOR-180 pulse with twice the rf-amplitude and half the pulse length $t_p$ for refocusing. In this case the frequency sweep is twice as fast as for the nominal SORDOR pulses and the quadratic phase should fully refocus to a normally phased spectrum.

In 2D experiments, the indirect dimension can be evolved as in conventional hard pulse experiments as the offset dependencies for excitation and back-transfer along $z$ compensate each other.

A disadvantage of the SORDOR pulses arises in heteronuclear experiments when pulses on different nuclei need to be applied simultaneously. In this case the same precautions apply that also must be taken i.e. for adiabatic or other shaped pulses (Khaneja, 2018; Foroozandeh, 2020). Simultaneously applied hard pulses should be aligned to either the left or right hand side

of the SORDOR pulses. Central application of pulses would need to be considered for a specific SORDOR pulse in use, e.g. with a toggling frame approach as demonstrated for selective pulses in INEPT-transfer elements (Haller et al., 2019; Ehni et al., 2021). Simultaneous application with other shaped pulses may also lead to unexpected effects, as coupling evolution might take place during the pulses (Ehni et al., 2021). Even if hard pulses are aligned left or right from the SORDOR pulses, offset-dependent coupling evolution cannot be avoided. However, in this case it is well-defined and in $^1H,^{13}C$-correlation experiment

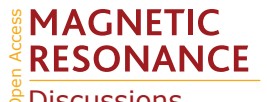

it can be used like in Chirp applications to compensate for offset-dependent coupling-size-compensation (Kupče and Freeman, 1997; Zwahlen et al., 1997).

It should be noticed that the proposed approach considers that resonance frequencies during delays stay the same. Exchange effects and radiation damping will lead to offset changes and because of the offset-dependent rotation axes finally to distorted phases. Such distortions may be undesired or even helpful in the identification of e.g. exchange processes.

Finally, we would like to make the remark that the relatively short SORDOR pulses used in this study are compensated for $\pm 5\%$ $B_1$-inhomogeneity with overall quality factors $\Phi > 0.9999$. This quality factor corresponds to a quite good performance. However, signal amplitudes in Fig. 4 reveal larger experimental modulations in pulse amplitudes than theoretically expected. To improve experimental performance, pulse shape optimisation eventually requires the adaptation of optimisation procedures with penalty functions regarding non-smooth waveforms (Goodwin and Kuprov, 2016; Goodwin, 2017) or non-adiabaticity

(Rosenfeld and Zur, 1996; Brif et al., 2014), which will be subject to future efforts.

## 5   Conclusion

A generalisation of the Böhlen and Bodenhausen concept by the use of matched SORDOR-90 and SORDOR-180 pulses is introduced. While SORDOR-180 pulses are equivalent to broadband inversion pulses with a defined phase behaviour, corresponding SORDOR-90 pulses as introduced in Goodwin et al. (2020) allow uniform $90°$ rotations around specific, quadratic

offset-dependent rotation axes. While the original Böhlen-Bodenhausen concept is limited to simple pulse-delay elements without mixing of states, the application of the concept using the SORDOR pulses also allows mixing with $90°$ elements, thereby enabling very broadband planar mixing and COSY-type experiments. As an example, we introduced a broadband [19]F-PROJECT experiment with very good results even for complex multiplets.

SORDOR-180 pulses afford the same rf-energy as time-optimal inversion pulses like BIP (Smith et al., 2001) or BIBOP

(Kobzar et al., 2004) pulses of the same quality. SORDOR-90 pulses require slightly more rf-energy than time-optimal BEBOP excitation pulses (Kobzar et al., 2004), but significantly less than corresponding BURBOP-90 time-optimal UR pulses (Kobzar et al., 2012). Therefore the SORDOR pulse pair represents the direct implementation of a broadband $90°/180°$ pulse-delay sequence with the least rf-energy deposition known so far, with a reduction factor of rf-requirements compared to conventional UR pulses slightly below 2. With this improvement in rf-usage we expect the introduced approach to be useful in a large

variety of experiments like fast pulsing 2Ds (see e.g. Kupče and Freeman, 2007; Schulze-Sünninghausen et al., 2014, 2017) and rf-limited imaging applications (Barker et al., 2001; Frankel et al., 2018).

*Author contributions.*   All authors contributed significantly to the work.



*Competing interests.* None of the authors of this paper has a financial or personal relationship with other people or organisations that could inappropriately influence or bias the content of the paper.

*Acknowledgements.* DLG thanks Martin Koos and Stella Slad for useful discussions on $^{19}$F-NMR and the PROJECT sequence. B.L. is grateful to Marcel Utz (Southampton) for intense discussions that finally led to the SORDOR project. He also thanks the HGF programme Information (43.35.02) and the Deutsche Forschungsgemeinschaft for financial support (LU 835/13-1).



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
