# Peer review of "SORDOR Pulses: Expansion of the Böhlen-Bodenhausen Scheme for Low-Power Broadband Magnetic Resonance"

_Magnetic Resonance, 2022_

## Author Response (AR1)

Dear Patrick,

Please find the detailed response to the referee's comments below. I uploaded the individual comments already specifically, but I included them here as well. Comments from both referees were indeed quite helpful and we learned quite a bit about our pulses, actually. We also went through the manuscript and polished the language.

There is one problem, though, that I hope will be tolerable: By accident I was deleting the whole directory with the original Latex-manuscript. It was entirely my stupidity (and of course no backup). Luckily the revised version survived, which was more luck than anything else. As I was the one with the .tex file and all others only made comments in the .pdf, I cannot easily provide a proper track-changes file – I would need to build up a latex file from a pdf version from scratch, which is a lot of work. I will therefore just submit the manuscript twice. As most changes are explicitly stated in the response to the referees, I hope this lapsus can be tolerated.

With this I can only hope that you will find the manuscript acceptable for publication.

Sincerely yours,

Burkhard (on behalf of all authors)

Referee #1, Philippe Pelupessy

Dear Philippe,

Thank you very much for the detailed reading and constructive comments. Please find in the following in which way we addressed your suggestions, which also involved the re-recording of quite a number of experiments. The manuscript certainly improved considerable with your comments. We are confident that you will now find it acceptable for publication.

Sincerely yours,

Jens D. Haller, David Goodwin, Burkhard Luy

This article presents the application of a class of broadband Universal Rotation pulses, so called SORDOR pulses. The particularity of these pulses is that the rotation axis depends quadratically on the offset frequency and that they have low RF power requirements. By matching the different pulses in a sequence, SORDOR pulses can be used in virtually any experiment (although sometimes quadratic phase-corrections need to be applied in the data processing). The power of the method is demonstrated on a very demanding "19F-project" experiment. The concept described in this article could have a very wide applicability (even if a, as the authors acknowledge, the pulses still need to be improved) and are of much interest for the readership of *Magnetic Resonance*. I recommend its publication after the following comments have been addressed:

- Figure 4. The profile of the echo sequence with hard pulses seems to be much broader than the one of the of the 90 degree pulse, probably because the coherence pathway selection is not ensured. Usually, it is more instructive to look at a sequence which ensures the correct coherence selection pathway (by gradients or EXORCYCLE for example).

>>> We very much thank you for this comment! The performance of the hard pulses in the original manuscript did not take the coherence pathway selection into account. We re-recorded corresponding experiments with gradient-based coherence selection (see new Figure 4) which much more demonstrates the advantages of SORDOR pulses compared to the hard pulse approach!

- Figure 4. The amplitudes for the SORDOR profiles vary quite a bit, which is exacerbated for the echo sequence. The authors should quantify these variations. What are the consequences for an experiment with multiple echos (like the project sequence)?

>>> By re-recording the offset profiles with gradient-based coherence order selection also the offset profiles turned out to have much less variation, actually corroborating theoretical offset profiles. We added values for upper relative variations in the text.

- Figure 5. "Clearly the relaxation of all signals can be easily followed using up to 128 perfect echoes". This figure on its own is not very instructive. Is the decay exponential (clearly not the case for the signal around -110ppm)? How much of the decay is due to imperfections of the pulses (see previous comment) or imperfect homonuclear decoupling? How close is the decay rate to the actual R2?

>>> We apologize that we did not show signal decays and also did not give any characterization of the PROJECT experiment. We re-recorded the spectra and included the decays for the 19F-PROJECT experiment in the SI. The signal decay for most signals is indeed exponential, only signals with expected 19F-19F second order artefacts deviate significantly, which is to be expected. We tried to compare the results with a hard pulse version of the 19F-PROJECT experiment for verification of exponential decay rates, but the hard pulses could not cover sufficiently the chemical shift ranges of coupled nuclei in the used sample. Resulting hard pulse PROJECT spectra resulted in highly distorted multiplets from which extraction of exponential decays was not at all possible. This made us aware that the PROJECT experiment in general does not lead to specific R2 rates of a single signal, but to relaxation rates determined by the inphase coherence transfer under planar mixing conditions of the full 19F spin systems. In addition, SORDOR pulses will lead to a (small) contribution of longitudinal relaxation. We rewrote parts of the experiment description accordingly. To test the performance and have a comparison with the hard pulse version of the PROJECT experiment, we measured the decay on a doped D2O sample and included the resulting exponential decays for the different versions in the SI.

- Line 162: "A disadvantage of the SORDOR pulses arises in heteronuclear experiments when pulses on different nuclei need to be applied simultaneously." This also applies for

homonuclear experiments. For example, does the fact that the coupled nuclei are not touched simultaneously has any effect on the perfect echo?

>>> The answer to this comment turns out to be quite complex. While adiabatic pulses due to the defined frequency sweep indeed show strong effects to coupled nuclei with considerable chemical shift difference, SORDOR pulses do not have an explicit frequency sweep, but a much more complicated trajectory during the pulse. An examination of the SORDOR pulses using average Hamiltonian theory shows (data not shown) that homonuclear coupling evolution is generally very low. Only nuclei with very small chemical shift differences close to the strong coupling limit will experience significant effects. We add a comment concerning strong coupling evolution within the context of the 19F-PROJECT experiment. In addition, we considerably changed Figure 3 showing now the quadratic phase behavior rather than the linear phase sweep, as your comment made us aware that this representation might be misleading.

-Sometimes the writing is a bit sloppy and imprecise (in particular in the introduction). For example: "With the advent of 1.2 GHz NMR spectrometers practically all common heteronuclear experiments require the use of amplitude- and phase-modulated pulses due to the enlarged bandwidths that need to be covered...". The need of broadband pulses has only to do with the bandwidths that need to be covered, not whether the experiment is heteronuclear or homonuclear. A very common 15N/1H HMQC (or even an HNCO) does probably not need shaped pulses even at very high fields, while many homonuclear 13C experiments do.

>>>Indeed, an HNCO experiment can be run with hard and selective shaped pulses. In case of the 1H,15N-HMQC, however, broadband pulses are utterly needed for most samples. Nucleic acids, for example, contain amino and imino nitrogens, making it impossible to cover all resonances in a single experiment using hard pulses. And even in well-behaved proteins the N-terminal NH2-group is usually not covered with hard pulses. We rephrased the wording to "… a large number of experiments requires …".

-Finally (this is more a comment to the editors of the journal, since the authors are not to blame): the format of the citations makes some parts of the article virtually unreadable (for a striking example see the second paragraph of the introduction). I think many readers would appreciate a change in format.

>>> We agree that citations using numbers would lead to less distraction when reading the text. But we leave any decision regarding the format to the Journal.

Reviewer #2

Dear Referee #2,

We also thank you very much for your knowledgeable comments, which certainly improved the readability of the manuscript significantly, particularly to readers that are less familiar with the details of pulse design. Please find the specific replies to your comments below. We very much hope that with the changes made you will agree to publishing the revised version of the manuscript.

Sincerely yours,

Jens D. Haller, David. L. Goodwin, Burkhard Luy

In this manuscript, Haller et al. describe a possible application of a recently reported class of radio-frequency pulses to low-power broadband NMR. The SORDOR pulses were designed with optimal-control theory to achieve universal rotation by 90° or 180° and a quadratic offset dependence of the phase. In this work combination or SORDOR pulses are used, with a specific example given for the case of the PROJECT pulses sequence. The idea of using OC-derived pulses for low power broadband MR, and, e.g., allow for broadband mixing, is interesting and useful. The reported experimental data is also well presented. However, several aspects of the theory and simulation are very difficult to follow for the non-expert reader. I recommend addressing the following issues before the paper can be considered for publication. They mainly consist in giving more background for the key concepts and results, and stating the main conclusions of the paper much earlier.

Throughout the paper, reference is made to the Böhlen-Bodenhausen scheme. It would be useful to explain what this actually is. My expectation was that this phrase referred to the use of combinations of frequency modulated pulses to remove the non-linear offset dependence of the phase that arise when using just one (the first paper by Böhlen and Bodenhausen). But the authors specifically do not show this. So what it the Böhlen-Bodenhausen scheme ? Using combination of frequency swept pulses ? Or is there some specific combination that should be met ?

In the same line, it might be useful to name the method with a phrase that describe what it does, rather than the name of those who first described it (especially since they described several different things).

>>> Böhlen and Bodenhausen published several different refocusing approaches based on linearly frequency-swept adiabatic pulses that result in quadratic phase behavior with respect to offset (as the phase is the first derivative of the frequency with time). We call the sum of all these refocusing approaches based on quadratic phase pulses the Böhlen-Bodenhausen concept. We explain this now in more detail in the introduction.

The authors write "the quadratic phase correction of the acquired FID may be compensated by using a SORDOR-180 pulse with twice the rf-amplitude and half the pulse length $t_p$ for refocusing. In this case the frequency sweep is twice as fast as for the nominal SORDOR pulses and the quadratic phase should fully refocus to a normally phased spectrum." Considering the relevance of having directly phasable spectra, the authors should demonstrate this experimentally…

>>> We had a close look at the compensation of the quadratic phase. It turns out that ideal conditions are obtained with a SORDOR-180 pulse with rf-amplitude and pulse length scaled

by square-root 2. As a result, no phase correction is needed with this type of refocusing before acquisition. We demonstrate the approach experimentally in new Figure 4 and changed/added to the text accordingly.

… especially since the introduction states "if a simple quadratic phase correction of the spectra can be tolerated". In which cases is there a solution if it cannot be tolerated ?

>>> We reformulated the sentence. In some cases quadratic phase corrections might lead to spectral artefacts. For example, if a very broad background is present in a spectrum a quadratic phase correction might result in a wavy baseline. But with the refocusing using the scaled SORDOR-180 pulse the quadratic phase correction can be avoided, circumventing also the background problem.

One of the main point seems to be that SORDOR pulses achieve universal rotation, which classic chirp pulses do not. Figure 1 illustrates this, but it is difficult to follow without a more detailed description of what "effective rotation" means, and what the different frames are (x, y, z) and (X, Y, Z). Please give more background information.

>>> We apologize for the confusion caused by using lower-case (x,y,z) in the main text and upper-case (X,Y,Z) in Figure 1. The coordinates in both cases refer to the rotating frame. We have changed Figure 1 to now include also lower-case (x,y,z). The effective rotation of a shaped pulse consisting of n piecewise constant individual pulses is defined via its effective propagator $U_{eff} = U_n \ldots U_1$, which for a single spin ½ represents a simple effective rotation in 3D space. We added the mathematical definition with a comment in the theory section.

It is known that linearly frequency swept pulses with gamma B1 / (2pi) = sqrt(BW/Tp) achieve uniform 90° excitation (see Tal and Frydman, Prog. Nucl. Magn. Reson. Spectrosc. 2010), with a quadratic dependence of the phase. How can this be reconciled with Fig. 1? And with the statement "it is obvious that the adiabatic pulse can only used to excite a single component". Addressing these two questions and the previous one would help (the non expert reader) to understand what is it that can be done with SORDOR pulses and not chirp pulses.

>>> First, the authors would like to make the statement of the referee more precise: linearly swept pulses (under certain conditions) achieve uniform excitation with an effective 90° flip angle starting form z-magnetization. The original z-magnetization is then distributed in the transverse plane with a quadratic dependence of the resulting x,y-phase with respect to the offset. A "uniform 90° excitation" as stated by the referee is therefore NOT a universal 90° rotation (UR90)! The effective flip-angles of initial x- or y-magnetization may and will deviate considerably from 90°! Only a single component, the initial z-magnetization, experiences a "uniform 90° excitation". It is therefore a classical PP (point-to-point) pulse with a quadratic phase behaviour. The adiabatic pulse can therefore not directly be used as a 90° mixing pulse, like in a COSY-type experiment.
With the SORDOR concept this direct mixing is possible without any addition. Next to a clarification in lines 33-38 (see below) we also introduced an explaining sentence in the conclusion.

Looking at Fig. S1, the difference between SORDOR and chirp pulses seems to be significant mostly for 90° pulses. It is stated in the conclusion but it would help if it were stated much earlier. It would be interesting to rearrange Fig. S1 so that it can be included as Fig. 1.

>>> We rearranged the previous Fig. S1 to be new Fig. 1. We state in the main text close to Fig. 1 regarding the similarity of SORDOR-180 and CA-Chirp-inv pulses: The SORDOR-180 pulse shows the very same rotation behaviour as the SORDOR-90 pulse, just with an effective 180° rotation angle in the x,y-plane throughout the optimised offset range. As such it resembles very much the behaviour of the CA-Chirp-inv pulse of same duration and rf-amplitude. However, the SORDOR-180 pulse is directly matched in its phase behaviour to the SORDOR-90 pulse, which for an adiabatic inversion pulse would only accidentally be the case.

It would help to address all of the above if l. 33-38 could be rewritten in more details:. what does "matching pulse shapes" mean ?
. "the inversion pulses acts as a refocusing […]"; how is that a consequence of the fact that "the effective phases of the pulses are matched"
. "Matching UR-90 pulses, however, […]"; this sentence seems to be the key part of the paper but it is difficult to understand.
. the explanation on COOP pulses is confusing, as it mentions "the least amount of restrictions" and then "similar restrictions".

>>> We have rewritten and extended lines 33-38 and the following paragraph to explain in more detail the matching of the pulse shapes: "Depending on the offset $\omega_z$, adiabatic excitation transforms z-magnetization into transverse magnetization with pulse-dependent phase angles $\alpha(\omega_z)$ with respect to the x-axis. A following matched adiabatic inversion pulse provides an effective rotation around the phase $\alpha(\omega_z)+\phi$ with either constant or linearly offset-dependent phase $\phi$. As the effective pulse phases are matched in this concept, the inversion pulses act as refocusing (UR-180) pulses with a quadratic offset dependence of the corresponding rotation axes, where the quadratic phase originates from the linear frequency sweeps of the adiabatic pulses. Adiabatic excitation pulses, however, are per se PP pulses and direct matching UR-90 pulses are not available. A more general concept without the restriction to linear frequency sweeps is the COOP concept introduced by Braun and Glaser … . But also here excitation pulse shapes introduced so far are PP pulses, resulting in limited applicability."

The author could define, maybe in the SI, what the "pulse performance" and "quality factor" are.

>>> (theoretical) pulse performance and quality factor have been used as synonyms, both referring to Phi as defined in new equation (1). We have added Phi to "pulse performance Phi" or "quality factor Phi" at several positions in the text for more clarity.

A comment following the posting or review 1: I entirely agree with reviewer 1 that the citation styles makes some parts of the manuscript very difficult to read. It would be much preferable to use numbered references.

>>> As this subject is a matter of taste, we leave any format change applied to the editors.